# Multiple Object Detection Based on Clustering and Deep Learning Methods

**DOI:** 10.3390/s20164424

**Published:** 2020-08-07

**Authors:** Huu Thu Nguyen, Eon-Ho Lee, Chul Hee Bae, Sejin Lee

**Affiliations:** 1Division of Mechanical Engineering, Kongju National University, Cheonan 31080, Korea; nguyenhuuthu.vmu@gmail.com (H.T.N.); Eonho@smail.kongju.ac.kr (E.-H.L.); better6573494@gmail.com (C.H.B.); 2Division of Mechanical and Automotive Engineering, Kongju National University, Cheonan 31080, Korea

**Keywords:** K-means clustering, DBSCAN, multiple object detection, LiDAR, underwater sonar images

## Abstract

Multiple object detection is challenging yet crucial in computer vision. In This study, owing to the negative effect of noise on multiple object detection, two clustering algorithms are used on both underwater sonar images and three-dimensional point cloud LiDAR data to study and improve the performance result. The outputs from using deep learning methods on both types of data are treated with K-Means clustering and density-based spatial clustering of applications with noise (DBSCAN) algorithms to remove outliers, detect and cluster meaningful data, and improve the result of multiple object detections. Results indicate the potential application of the proposed method in the fields of object detection, autonomous driving system, and so forth.

## 1. Introduction

The detection of multiple objects is crucial in image processing. It has been investigated extensively owing to its potential wide application in numerous fields, such as computer vision, machine inspection, manufacturing industry, and self-driving cars. In This study, two scenarios were conducted using several deep learning and clustering algorithms on different data types, that is, underwater sonar image data for multiple submerged human bodies and object detection, and three-dimensional (3D) LiDAR data for multiple object classification and segmentation in an urban structure to be applied in autonomous driving. The fully convolutional network (FCN) [1] is used for underwater sonar data to detect the presence of the human body and other objects in the underwater environment and to segment those objects. The 3D point cloud LiDAR data of the urban environment are first subjected to a spherical signature descriptor (SSD) [2] to change spatial data to color image data; subsequently, the convolutional neural network (CNN) [3] is used on the output data to classify and segment objects in the environment. The outputs of both sonar and LiDAR data have a high amount of noise, which result in the low performance results of detection and segmentation tasks. Hence, the two scenarios above were managed using two clustering algorithms, that is, K-Means clustering [4] and DBSCAN [5] to detect and remove noise such that the results are improved and optimized; the flow chart of the proposed method is shown in Figure 1. The main goal of This study is to achieve multiple object detection by applying K-Means clustering and DBSCAN algorithms. Segmented results comprising underwater sonar images and 3D point cloud LiDAR data obtained from the authors’ previous studies were used to remove outliers, whereas object detection was performed using the proposed method.

## 2. Related Work

Although interesting, the underwater environment is associated with many challenges, such as limitations in view sight and difficulties pertaining to direct approaches by human beings. Owing to the unlimited promising opportunity and potential application of the underwater environment, studies regarding the latter have increased recently [6,7,8]. K. J. DeMarco’s proposed a diver detection and tracking method that applied the hidden Markov model for cluster classification using forward-looking sonar data [9]. In Reference [10], side-scan sonar data were used for target detection, where a single threshold algorithm was applied to remove shadows in sonar images to improve detection results. The authors exploited the integral-image representation to apply a forward-looking sonar imagery on an autonomous underwater vehicle to perform object detection in real time [11]. The authors improved a convolutional neural network (CNN)-based classification model using underwater sonar imagery data. The proposed method using transfer learning was successful even when low-quality sonar images were used in object recognition [12]. In 2019, Matias Valdenegro-Toro proposed a method using a fully convolutional neural network on a Forward-Looking sonar image for regressing an objectness value that is challenging in object detection using sonar images. This method is expected to be application in object detection techniques on sonar images in the future [13].

Among the different data types and technologies that are currently being studied and applied for autonomous driving, such as vision camera, sonar sensor, radar sensor, GPS, and LiDAR sensor, LiDAR data are used the most owing to their advantages compared with the others. A vision camera might not operate in a weak light condition, whereas a sonar sensor is primarily used for close object measurements and detections with a close effect range. Radar data require a high and stable network connection, which is affected by weather conditions, real time positions, and so forth. Therefore, the results might not be accurate in some cases. Although LiDAR data have their own limitations, they can be collected quickly with high accuracy, operate well during the day and at night, and can be integrated with other data sources. Many studies on LiDAR data are being conducted to enhance object detection and segmentation to complete autonomous driving systems. In 2009, F. Moosmann et al. investigated non-flat urban environments for 3D LiDAR data. The authors considered transforming LiDAR data into a two-dimensional (2D) graph. The proposed graph-based segmentation method based on the concept of local convexity resulted in the excellent segmentation of non-flat ground and typical obstacles, such as pedestrians, cyclists, cars, and tree trunks [14]. In 2010, an efficient graph-theoretical algorithm model that operates on the combination of both the color information from a view camera and the 3D LiDAR point cloud from a laser sensor was introduced. The study enhanced the detection of segment boundaries both in indoor and outdoor environments, and the results were better than those of other studies pertaining to only vision or laser data [15]. In 2011, a set of segmentation methods for different 3D point cloud data types were proposed. Both dense and coarse LiDAR data were used with different segmentation methods, such as the cluster-All method and the mesh-based technique. The study results performed well on the segmentation of cars, trees, pedestrians, and so forth [16]. CNN-based object segmentation for LiDAR data was conducted on the large-area LiDAR data of New York city and indicated good results even with missing points in vehicle object detection and segmentation [17]. In 2019, Shaoshuai Shi et al. proposed the PointRCNN model that generate and detect objects used 3D point cloud data, This work conclude two stages of bottom-up 3D generation from the point cloud data and refines the 3D proposals in the canonical coordinate. The authors also used KITTI data for their work and received good result on 3D detection [18]. Qingdong He et al. worked on Sparse Voxel-Graph Attention Network (SVGANet) used KITTI raw data of 3D LiDAR point cloud that generates the local complete graph within 3D spherical voxel and global KNN graph through All voxels. By doing these, the sparcial features are extracted for the object detection in using 3D point cloud data [19].

Two clustering algorithms were used in This study to find and remove outliers in the input data of underwater sonar data and LiDAR data to improve the performance of multiple object detections. Section 3 introduces both the deep learning methods and clustering algorithms that were used to prepare the input data to achieve the study goal. Section 4 describes the experimental data preparation and the results. Finally, discussions and conclusions are provided in Section 5.

## 3. Materials and Methods

### 3.1. Deep Learning

#### 3.1.1. Convolutional Neural Network

As one of the main tools in computer vision, CNNs are used extensively for image recognition, image classification, object detections, face recognition, and so forth [3]. The CNN model is applied to image data, and the input images are processed and classified under categories such as animal, human, or other objects. Unlike human beings, computers view images in terms of numbers. All input images are changed from colors to numbers ranging from 0 to 255; therefore, the input data of images are changed to a matrix of numbers, with each number presenting a color state. A simple CNN model such as that shown in Figure 2 comprises an input and an output layer, although more hidden layers can exist between them depending on the type and complexity of the input data as well as the task requirement. The hidden layers of a CNN typically comprises a series of convolutional layers followed by the activation function layer, which is typically a ReLU function and followed by additional convolutions, such as pooling and normalization layers for optimizing the model. The end of the model comprises a flattened layer and some fully connected layers; these layers are followed by pooling and normalization layers as well. Activation functions that are typically used in the CNN model include the sigmoid function, the ReLU, and some other functions. Object recognition is realized by traversing All the layers above in the CNN model.

#### 3.1.2. Fully Convolutional Network

The convolutional neural network (CNN) structure typically comprises multiple convolutional layers interlaced with pooling layers and some fully connected layers at the end of the structure [20]. These layers apply filters, extract features from the input images, and render the output of convolution networks translationally invariant. The FCN model is an improved version of the CNN and is a class of deep neural networks. It is typically applied for analyzing visual imagery; an example of the FCN model is shown in Figure 3. The FCN network transforms All the fully connected layers into convolutional layers. It reduces the output size using the pooling operation while maintaining the input information [1]. With the extensive application of FCNs for image segmentation, the FCN model was used in This study to extract multiple objects out of the background in underwater sonar images to prepare the input data for the clustering algorithms.

### 3.2. Clustering

#### 3.2.1. K-Means Clustering

The idea of K-means clustering algorithms was first proposed by Hugo Steinhaus in 1956. However, the standard algorithm was first proposed by Stuart Lloyd in 1957 and was officially published in 1982 [21]. The K-means clustering algorithm is a method of vector quantization that aims to separate a number of initial input data into K clusters, in which each data point belongs to their specific group with the closest mean of that group. This algorithm is a simple yet popular and useful unsupervised machine learning algorithm. The letter “K” in the name of the algorithm represent the number of clusters in each dataset. This number is either predefined manually by users for each dataset based on the dataset and the requirement of each study or automatically defined based on some techniques, such as the elbow method or the Silhouette analysis method [22] after analyzing the dataset.

The data learning process is shown in Figure 4; a suitable K must be determined initially at the beginning. After defining K clusters, new centroids are selected. The K-means algorithm randomly selects K centroids, which are used as the beginning center points for every cluster in the dataset. Because This step is performed randomly, those points can either be the same data in the dataset or different ones.

In the second step, K groups are created in the dataset based on the new randomly selected centroids. Subsequently, the distances of All the points in the dataset to those k points are calculated, and each data point is grouped into the cluster of each centroid with the smallest distance. This calculation is performed using Equation (Equation 1), where $\left|\right|x_{i}-v_{j}\left|\right|$ the Euclidean distance between $x_{i}$ and $v_{j}$, c the number of cluster centers, and the $c_{i}$ is the number of data points in the *i*th cluster. By comparing these distances, All the points in the data are defined to their own cluster. After calculating All the data points in the dataset, K groups are formed with the respective member points.

The third step involves calculating the mean of each group by Equation (Equation 2) to obtain the new centers for All K groups. These new values typically differ from the initially selected ones. Subsequently, the algorithms will repeat the second step to define the new K group depending on the new centroids and the distance from All the points in the dataset to those center points. In the last step, the third step is repeated to obtain the new centroids for the K groups of data points. K-means clustering will stop when no changes occur in the new centroids, which means that the values of the previous clustering are the same as those of the new ones. At This step, the algorithm has finally clustered All data points into K different groups of data points with the same characteristics.
(1)$$j\left(v\right)=\sum_{i-1}^{c}\sum_{j=1}^{c_{j}}{\left(||x_{i}-v_{j}||\right)}^{2}$$
(2)$$v_{k}=\frac{1}{|c_{k}|}\sum_{i\in c_{i}}x_{i}.$$

#### 3.2.2. DBSCAN

In 1972, a closely related algorithm was introduces by Robert F. Ling [23]. Later in 1996, M. Ester published his work in the paper named “A Density-based spatial clustering of applications with noise“ (DBSCAN) which is then become a very popular and common clustering algorithm [24]. It is a well-known data clustering algorithm or a relevant algorithm that is typically used in data mining and machine learning. The input data are a set of All random and different points from the data source. DBSCAN will cluster the points that are close to each other in a specific group based on the distance between those points (the Euclidean distance are typically considered) and a preselected minimum number of points in one area. All the points that do not satisfy the two values above are marked as outliers or noise, which are points in low-density regions. The two elements mentioned above are two parameters required in the DBSCAN algorithm, that is, the eps and minpoints. They are selected based on analyzing the initial data. By selecting the appropriate values for these parameters, DBSCAN will cluster the input dataset into three different types of groups: the core, border, and outlier points.

The first parameter of the DBSCAN algorithm is the eps, which is a specific distance from an original data point to surrounding points. This value is set the same for All the points in the dataset to cluster them in their suitable clusters. If the distance between two different points is lower than or equal to the eps value, these points are considered as adjacent to each other. To ensure that the DBSCAN algorithms perform well, the dataset must be analyzed; subsequently, the optimal eps value for that specific dataset must be defined. This value must be set for each different dataset. If the eps value is extremely small, then a large portion of the dataset will not be clustered into a meaningful group but considered as outliers as it does not satisfy the number of points in a specific area to define the dense region. By contrast, an extremely high eps value will result in most of the valuable and meaningless data be merged into the same cluster. The K-distance graph of the dataset can be used to analyze and define the eps value. In general, a smAll eps value is preferable.

The second parameter is the minpoints, which is the minimum integer number of the points in one circle defined by the center as the original point, and the radius is equivalent to the eps. If there are more points in that circle than the minpoints, then those points are considered as part of a cluster. For datasets with noise, the greater number will typically yield better results in the more significantly meaningful clusters. Although three is the minimum value of the minpoints, This value should be greater when addressing a larger dataset.

The DBSCAN algorithm can be expressed as follows: From an original point, a circle is defined with the radius equivalent to the eps, and the number of points in This circle is calculated by comparing the Euclidean distance between the original point and All other surrounding points. The points in the dataset will be analyzed and calculated to define the point type. It will be considered in the group of the core points if the number of points in that circle with the eps radius is equal to or greater than the minpoints. For example, in the Figure 5, the minpoints was set to 3, so from the original point, if there are more 3 point in the circle area with the eps diameter including the original point, then that original point will be the core point. The core points are represented in black in Figure 5. If the number of points inside the circle is smaller than the minpoints and more than 1 point, then the original points will be considered as border points, which are represented in yellow in Figure 5. The remaining points that do not satisfy the two requirements above regarding the eps and minpoints which means there are no other points in the circle from the original point with eps diameterwill be considered as outliers or noise points, which are shown in red in Figure 5.

#### 3.2.3. Silhouette Analysis

Silhouette analysis is a method to evaluate the separation distance between the resulting clusters. The range of the silhouette value is [−1, 1]. By analyzing the data and providing the silhouette value on the graph, This method shows the similarity of one object or data point to its own cluster compared with other clusters in the dataset. The higher silhouette value indicates that the object is more likely to belong to its own cluster, whereas it is more separated to the other neighboring clusters. This method provides an insight into the selection of the optimal K value, which is the most important parameter in K-means clustering. In Equations (Equation 3)–(Equation 5), $C_{i}$ is cluster *i*, *d*(*i*,*j*) the distance between points *i* and *j* in cluster $C_{i}$, and *s*(*i*) the silhouette value. Figure 6 shows an example of using the silhouette method for K-means clustering on a dataset with two clusters. The method indicates that the optimal K value is 2 when the silhouette value is 0.72; This value is verified as shown by the dotted red vertical line.
(3)$$c\left(i\right)=\frac{1}{|C_{i}|-1}\sum_{j\in C_{i},i\neq j}d(i,j)$$
(4)$$b\left(i\right)=\min_{k\neq i}\frac{1}{C_{k}}\sum_{j\in C_{k}}d(i,j)$$
(5)$$s\left(i\right)=\frac{b(i)-a(i)}{\max\left\{a(i),b(i)\right\}}.$$

## 4. Experiment Result

### 4.1. Data Preparation

In This study, two different types of data were used: underwater sonar images and urban 3D cloud point LiDAR data. To collect the sonar data, results from previous studies were used by applying the FCN model on the underwater sonar images. Multiple objects, such as body mannequins, car tires, plastic boxes, and walls were extracted from the underwater sonar images. In This study, the FCN model was used to train and test the segmentation on the training and testing underwater sonar images. A set of 331 training images that were manually hand-labeled with labels of “body”, “tire”, “plastic box”, and “wall” were prepared for the training. After training with 331 hand-labeled images, 63 testing images were tested using the trained model to extract the segmented body and other objects including the box, car, tire, and wall, which were extracted from the background. Subsequently, these segmented body images were transformed to pixel data that were represented as a matrix comprising two values x and y in the coordinating elevation position of All plots of those objects in the segmented image, as shown in Figure 7. The data of the segmented objects in the underwater sonar images were then used for clustering algorithms to define the number of objects and detect their positions in the images.

Two different datasets of the 3D point cloud LiDAR data were used in the study. The first one was collected inside the KNU campus with a Velodyne 16 channel LiDAR sensor mounted on the top of an electronic vehicle; This dataset was named KNU. The second set was from the KITTI LiDAR raw data [25]. Both of these datasets were subjected to SSD to change the data type from 3D point cloud data to image data, which represent different objects and contain the properties of the 3D data. Subsequently, these image data were used with the CNN model and then classified into five different labels, including people, tree, car, wall, and floor. In This study, the simple CNN structure was used with the repeated convolution layers and the maximum pooling layers, The image input data size was 28×28 pixels. In the former layers, the kernel size were 5×5 while in the latter layers, the size of the kernel filter were 3×3. The activation function were used in the CNN model are ReLU function, and the softmax function in the last layer. By applying SSD and the CNN model, the input 3D point cloud LiDAR data can be classified into five different labels. The proposed method for DBSCAN algorithms should be applicable to two-dimensional data. Therefore, the initially labeled LiDAR data must be transformed from 3D to 2D. When changing the data, some of the data points were lost owing to the overlap of the z-coordinate; therefore, the corresponding number of data points must be added according to those missing parts. Figure 8 shows an example of the 2D input LiDAR data.

### 4.2. Experiment Result

In This study, 63 segmented images of underwater sonar images were used; they were used in two different scenarios based on the characteristics of the dataset. In the first case, the background was removed from All images, whereas in the second case, the clustering algorithms are used with data that include the objects and the background. First, K-means algorithms were used on the input data to classify the multiple objects in the images. Because K-means requires defining the initial “K” value, which presents the number of clusters, the authors used two different methods to define it: the elbow method and the silhouette analysis. The elbow method shows the relationship between the sum of the squared distance of the dataset and the cluster k, as shown in Figure 9a. By defining the characteristics of the graph, the number K was selected at the point when the graph changed significantly, where the optimal K value was shown. For example, in Figure 9a, the optimal K value was 5; at This point, the direction change in the graph was the most significant. Silhouette analysis was also performed to define the K value for the underwater sonar data. This method is used to show the separation distance of All clusters in the data with different K values. The K with the highest silhouette value will be selected. As shown in Figure 9b, the number of clusters was 5 when the highest silhouette value was 0.91.

Figure 10c shows the successful clustering result of multiple objects in the underwater images after using the K-means clustering algorithm. As shown in the image, five clusters are represented in different colors and bounded by red boxes. Furthermore, DBSCAN was used to cluster the sonar data in This study. Figure 10d shows the result of clustering multiple objects in the images into different colors and bounding each cluster with a red box.

Table 1 and Figure 11 show the clustering results on the 63 underwater sonar images. By applying two clustering algorithms, it was shown that the K-means clustering accuracy was lower compared with that of the DBSCAN, which was higher than 90% in the two different datasets with and without the background. The silhouette analysis for defining the K value indicated better results than those of the elbow method, with the highest accuracy of 84.13% recorded on the dataset without the background. In two different dataset, the dataset with the background indicated better results because the data points were more concentrated and had less noise.

The 2D segmented LiDAR results from the 3D LiDAR input data indicated two different types. The data were separated into different sets according to their types, and each set comprised 12 different images. The aim of This separation was to test and compare the accuracy between the different datasets and different data types such that the parameters can be modified to improve the quality of the clustering result. The first one was collected in the KNU campus, and it is known as KNU. This set comprises two smAll subsets, with each subset comprising 15 images. All the data were recorded inside the KNU campus in the normal weather. In This dataset, the number of people in each image differed; in some images, people were not present but more outlier points appeared. The second set was obtained from the KITTI 3D raw LiDAR data; This dataset comprised five subsets, and each subset comprised 12 different images. There were more people in This dataset and they were present in All the images. Although these images showed the data of multiple real people, they contained many outliers; therefore, the author proposed a method using DBSCAN clustering to remove All of those outliers and detect the presence and position of the people data. All the image data were used for the CNN model are the individual images of car, people, floor, tree, building. Though the clustering algorithms for multiple object detection used on the total 3D point cloud LiDAR data to cluster multiple objects by analysis the distribution of the data. All the data for clustering step was randomly chosen from the KITTI dataset, the data in All subsets were also randomly selected so the used data in This study at the clustering step have presented the generality of the experiment on the LiDAR data.

In Figure 12, the distance relationship between nearest neighbors and the distribution of data points in both the KNU and KITTI datasets are shown in graphs a and b for the sample image c. When the elbow method was used on the K nearest neighbors, the graph was used to define the eps value. This process was performed using All LiDAR data in both the KNU and KITTI sets. The significant difference between the data of the body and outliers is that those of the body had a much higher density compared with those of the outliers. As shown in Figure 12b, the data distribution values in the body area were higher than 40. The minpoints required by DBSCAN was selected based on this value.

The DBSCAN clustering results are shown in Figure 13, where the left figure a shows the LiDAR input data that include All objects, such as car, people, and tree. The middle figure contains only the information of people in the input LiDAR data, whereas the right figure shows the DBSCAN clustering result, in which people were marked in different colors and stored inside the red boxes. All the remaining outlier points are shown in blue; these data points are outside the red boxes and they represent the noise or people mislabeled by the deep learning algorithms.

The DBSCAN model successfully performed the definition on both the KNU and KITTI datasets regardless of whether people existed in All the images. As shown in Table 2 and Figure 14, the number of people resulting from the clustering results in each subset is shown in the “total people” row, as mentioned above, whereas the errors of mistyping noise points as people are shown in row 5. In some cases involving both the KNU and KITTI LiDAR data, the clustering of outlier points into people type failed owing to their features, which were similar to those of the real people data. In All cases, the clustering accuracy was approximately 90%. However, the accuracy of clustering the people data was 100%, in which outliers in three among All seven datasets were removed. The number of outlier points removed from the input data using the proposed method is shown in the final row; there were hundreds of points, which depended on each small subset.

## 5. Discussion and Conclusions

Underwater sonar data and 3D point cloud LiDAR data were investigated using clustering algorithms to remove outliers in the input data and define the presence of multiple objects. Both K-means clustering and DBSCAN algorithms demonstrated good results on two different datasets, with the highest accuracy of 100% when using DBSCAN on the LiDAR data. The results of This study may be applicable for detecting multiple objects in both the underwater and terrestrial environments to study underwater sciences and autonomous driving systems, respectively.

## Figures and Tables

**Figure 1 sensors-20-04424-f001:**
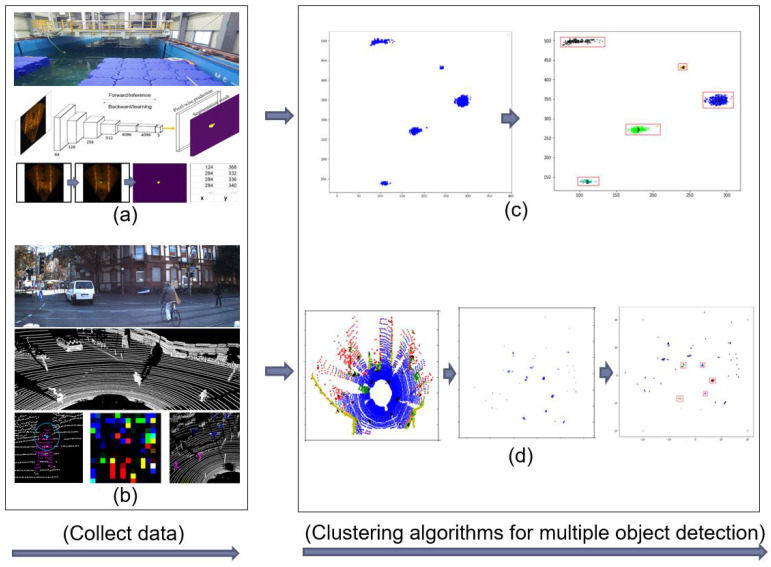
Flow chart of operation: (**a**) Collecting segmented objects using FCN model, (**b**) Collecting segmented human body using spherical signature descriptor (SSD) and convolutional neural network (CNN) model, (**c**) Multiple object detection using clustering algorithms on underwater sonar data, (**d**) Multiple human body detection using clustering algorithms on urban 3D point cloud LiDAR data.

**Figure 2 sensors-20-04424-f002:**
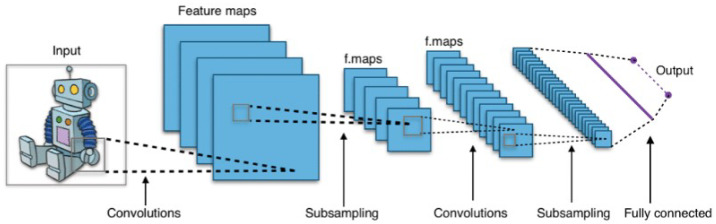
Structure of CNN model.

**Figure 3 sensors-20-04424-f003:**
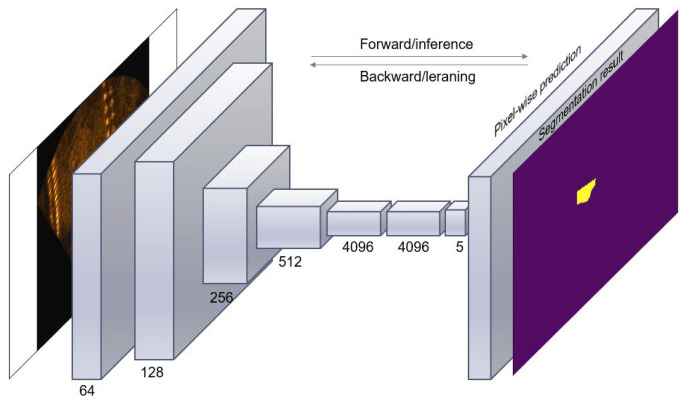
Structure of FCN model.

**Figure 4 sensors-20-04424-f004:**
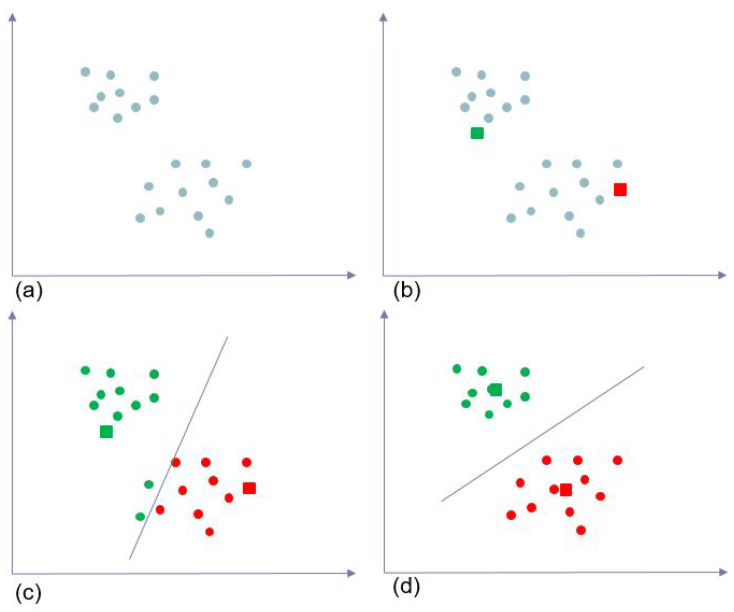
K-means clustering algorithm: (**a**) Initial input data; (**b**) Randomly select centroids; (**c**) Cluster input data into K clusters; (**d**) Obtain new centroids.

**Figure 5 sensors-20-04424-f005:**
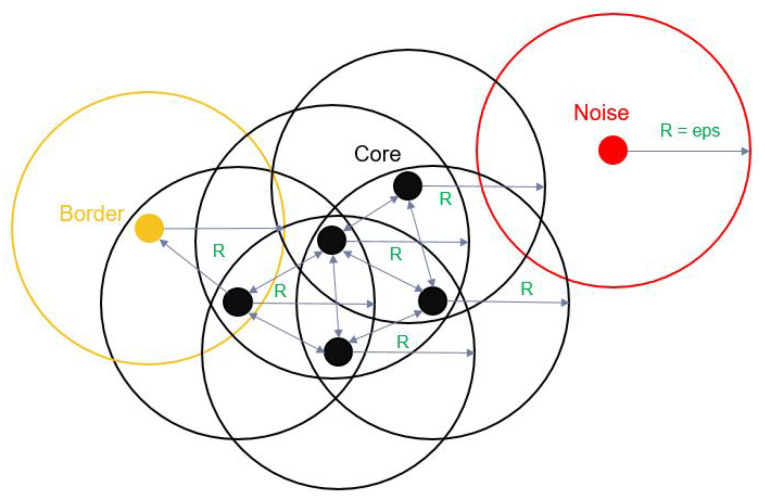
DBSCAN algorithm.

**Figure 6 sensors-20-04424-f006:**
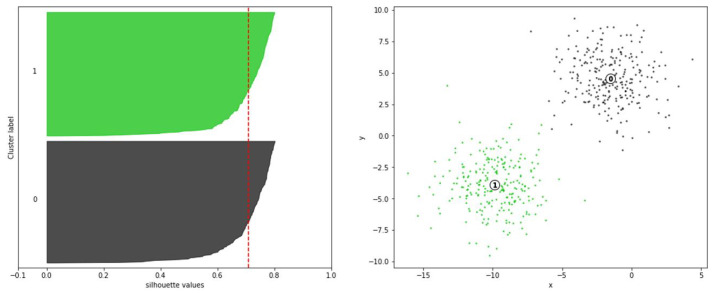
Silhouette graph for defining K value of K-Means clustering.

**Figure 7 sensors-20-04424-f007:**
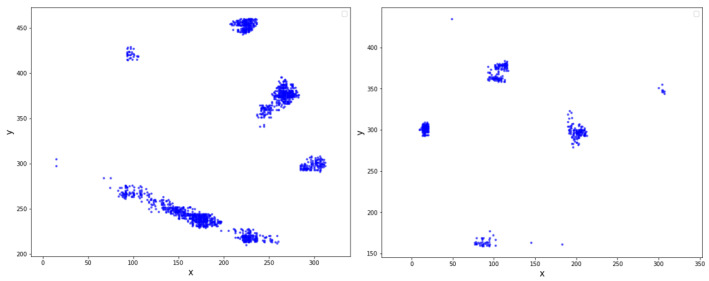
Sample input data of segmented sonar object.

**Figure 8 sensors-20-04424-f008:**
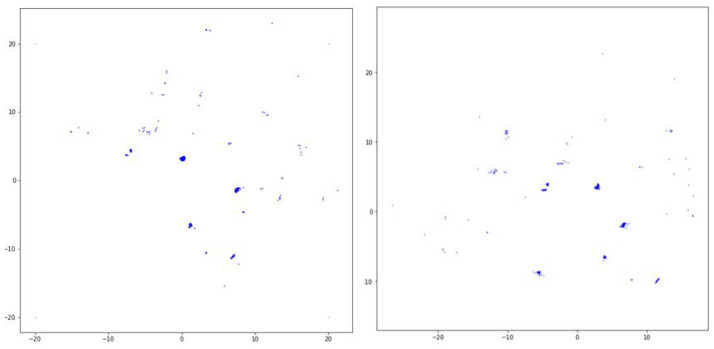
Sample input data of segmented LiDAR data.

**Figure 9 sensors-20-04424-f009:**
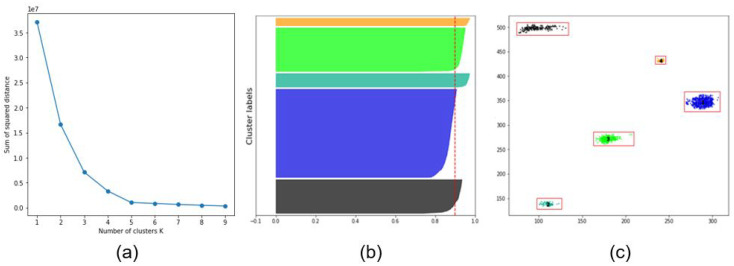
Elbow and silhouette analysis methods for determining K values of K-means clustering: (**a**) Elbow method; (**b**) Silhouette analysis graph; (**c**) Example of clustering result.

**Figure 10 sensors-20-04424-f010:**
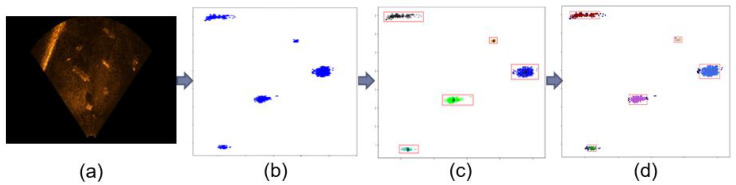
K-Means and DBSCAN result on Sonar data: (**a**) Real underwater sonar images, (**b**) Segmented images, (**c**) Clustering result using K-Means clustering, (**d**) Clustering result using DBSCAN.

**Figure 11 sensors-20-04424-f011:**
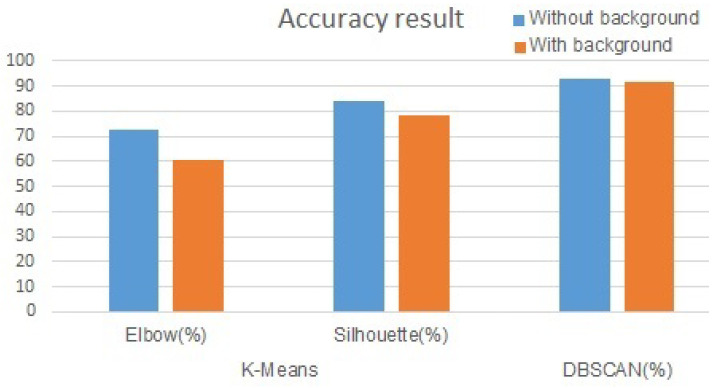
Clustering result graph of Sonar data.

**Figure 12 sensors-20-04424-f012:**
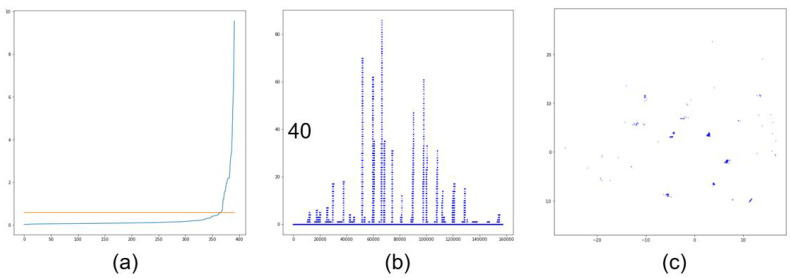
K nearest neighbors; distribution of data points in LiDAR data: (**a**) Elbow method graph, (**b**) Distribution graph, (**c**) Example of LiDAR input data.

**Figure 13 sensors-20-04424-f013:**
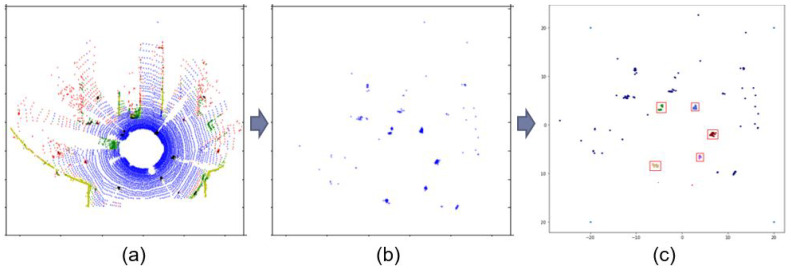
DBSCAN result on LiDAR data: (**a**) Initial input LiDAR data; (**b**) Segmented LiDAR data of people; (**c**) Clustering result of LiDAR data.

**Figure 14 sensors-20-04424-f014:**
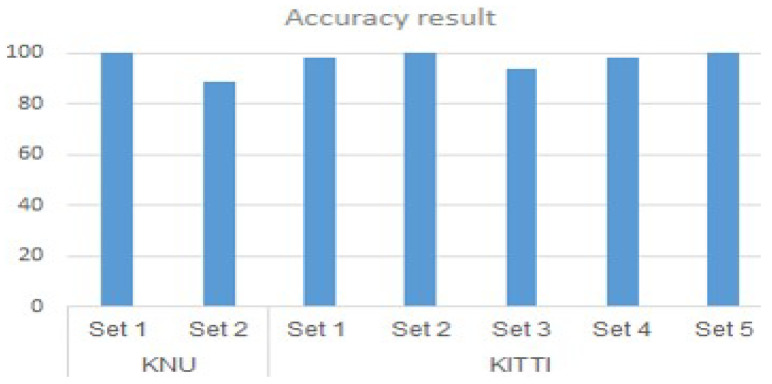
The clustering result graph on LiDAR data.

**Table 1 sensors-20-04424-t001:** K-Means clustering and DBSCAN algorithms result on sonar data.

Type	K-Means Clustering		DBSCAN(%)
	Elbow(%)	Silhouette(%)	
Objects without background	72.61	84.13	92.86
With background	60.84	78.57	91.67

**Table 2 sensors-20-04424-t002:** DBSCAN clustering result on LiDAR data.

Type	KNU		KITTI				
Data Set	1	2	1	2	3	4	5
Number of images	15	15	12	12	12	12	12
Total people	18	16	56	52	47	54	63
Mistype of people	0	2	1	0	3	1	0
Accuracy (%)	100	88.89	98.25	100	94	98.18	100
Outlier points	56	95	354	284	367	349	233

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
