# Peer review of "Multiple Object Detection Based on Clustering and Deep Learning Methods"

_sensors, 2020, doi:10.3390/s20164424_

Round 1
Author Response
Dear Reviewer,
Please take a look at the file that we sent to you for our revised manuscript based on the reviewer's comments.

Reviewer 2 Report
The paper titled "Multiple object detection based on Clustering and Deep Learning" attempts to detect multiple objects using the techniques of Deep learning and clustering. In my views the paper in its current version is not suitable for publication. I have the following observations:
1) This paper is week in terms of the contribution. Authors are advised to incorporate more research in this work.
2) The research problem is not explained properly, more emphasis is given on techniques like deep learning and clustering. Novelty of this paper if not explained properly.
3) Picture resolution for figures 3, 4 and 5 should be improved.
4) The discussions on clustering needs to be improved, specifically the DBSCAN algorithm is not covered properly. The fact that it is a density based clustering algorithm is missing in the text. Further, the Figure 5 does not explain why are some points marked as core, border and noise? In order to explain these concepts the authors should specify what is the minpoints value for this picture.
5) There is scope of language improvement also.
Author Response
Dear Reviewer,
It is the authors' pleasure to revise our manuscript based on the reviewer's comments so that our paper could be more improved. Please take a look at the file that we sent to you.
Thank you,
The authors

Reviewer 3 Report
This article focuses on the rapidly growing and popular topic of object detection using deep learning methods. The authors also consider the solution to this task on their own data of an underwater sonar.
However, the article has a number of significant comments:
1) Sections “1. Introduction” and “2. Related work” does not contain links to modern sources of 2019-2020. The mentioned modern works [2], [6], [7] belong to the authors of the article. At the same time, there are a large number of recent scientific publications on the subject of multiple object detection in 2D images and/or 3D LiDAR data. Authors should analyze contemporary works, for example, collected on the online resource https://paperswithcode.com/area/computer-vision/object-detection.
2) There is no novelty of the proposed multiple object detection approach based on clustering and application of neural networks: the described clustering methods K-Means and DBSCAN are well-known (it is not clear why their detailed description is given in the article), the described CNN and FCN models are also well-known (respectively for classification and data segmentation).
3) There is no comparison of the methods proposed in the article with state-of-the-art-approaches for solving the object detection problem.
4) There is no justification for the choice of deep neural network architectures based on the modern state of research. The details of their implementation are not indicated (sizes of convolution kernels, strides, sizes of feature map tensors, etc.), as well as parameters of their training.
5) It is unclear why the authors use the accuracy metric when evaluating the quality of the solution to the detection task, while the average precision (AP) is a common metric.
6) It is strange why only 60 images were selected from the KITTI dataset to analyze the results of object clustering (detection), while this dataset contains much more images. Using a complete dataset would probably significantly reduce the high quality metrics of the proposed approach on it.
7) Figure 14 duplicates the data shown in table 2.
8) In the formulas (1) and (2) some indices “i” and “j” are confused.
9) Typo on page 9: “two different dataset”.
Author Response
Dear Reviewer,
The authors thanks to the reviewer's comments for those the authors could improve the manuscript more.
Please take a look at the file that we sent to you.
Thank you
The authors

Round 2
Reviewer 3 Report
The authors took into account the main remarks of the reviewer, it remains to correct the following small points:
1) the text does not explain what j(v) is in formula (1). In addition, in formula (1) it is necessary to correct the summation limits. Most likely, the first sum should contain at the bottom i = 1, and the second sum at the top should have c_i. Also in the article there is no explicit explanation of the notation x_i and v_j, this should be added.
2) in formula (2) there is an error in the notation of summation, apparently it should be like this: i ∈ c_k.
3) in figure 4 and figure 12 the authors should specify the labels of the axes.
4) Duplication of Figure 14 and Table 2 should be eliminated, this information is superfluous.
It should be added that the lack of comparison of the methods proposed in the article with state-of-the-art-approaches for solving the object detection problem significantly complicates the assessment of the practical significance of the results obtained by the authors of the article.